# Design of Nature Tourism Route in Chimborazo Wildlife Reserve, Ecuador

**DOI:** 10.3390/ijerph18105293

**Published:** 2021-05-16

**Authors:** Danny Castillo-Vizuete, Alex Gavilanes-Montoya, Carlos Chávez-Velásquez, Paúl Benalcázar-Vergara, Carlos Mestanza-Ramón

**Affiliations:** 1Instituto de Investigaciones (IDI), Proyecto de Investigación Diseño de Productos Turísticos (RETOUR), Escuela Superior Politécnica de Chimborazo, Panamericana Sur km 1½, Riobamba EC-060155, Ecuador; renato.chavez@espoch.edu.ec; 2Faculty of Natural Resources, Escuela Superior Politécnica de Chimborazo (ESPOCH), Panamericana Sur km 1½, Riobamba EC-060155, Ecuador; 3Natural Resources Management Faculty, Lakehead University, Oliver Road, Thunder Bay, ON P7B 5E1, Canada; pbenalca@lakeheadu.ca; 4Research Group YASUNI-SDC, Escuela Superior Politécnica de Chimborazo, Sede Orellana, El Coca EC-220001, Ecuador; 5Departamento Economía Financiera y Dirección de Operaciones, Universidad de Sevilla, 41018 Sevilla, Spain; 6Instituto Superior Tecnológico, Universitario Oriente, La Joya de los Sachas EC-220101, Ecuador

**Keywords:** cycling, hiking, multiple criteria, GIS designing, cost distance, tourism planning

## Abstract

The design of new routes is a specific strategy to improve tourism management and to increase the attractiveness of landscape features, promoting activities as a part of sustainable development. This study proposes the design of alternative multi-parameter tourist routes in the Chimborazo Wildlife Reserve based on spatial network analysis implemented in ArcGIS 10.5^®^ software. Tourist interest points were identified and mapped using spatial analysis software, then two routes for bicycles and hiking were defined as being the most efficient, based on the most frequented tourist attractions. The main contribution of this study is the identification of optimal routes for vehicular, bicycling, and hiking traffic through tourist attractions, considering variables such as the time, distance, average circulation speed, road state, and tourist facilities. As a result, two routes were identified. Route one includes 17 tourist attractions, five lodging establishments, four food centers, and one health center. On the other hand, route two includes 11 tourist attractions, two lodging and food establishments, and one health center. The final contribution of this research is to maximize tour satisfaction by presenting new routes of visiting tourist attractions due to the growing demand in the Chimborazo Reserve.

## 1. Introduction

The tourism industry showed uninterrupted growth after the 2009 financial crisis, but by 2020, with the arrival of the COVID-19 pandemic, tourism has recorded its worst year ever with a 74% drop in international arrivals [1]. This industry has been one of the main pillars of social, environmental, and economic development worldwide [2,3]. As such, the tourism sector is continually evolving [4], with the today’s tourists seeking for and documenting their emotional experiences and travel memories by the use of technology [5] instead of just simply looking [6,7]. In addition, the new tourist trend is that of autonomous search for information [8], often resulting in combining leisure activities, sports, and relaxation in relation to a given tourist location [9,10,11,12]. Considering that experience is the essence of tourism [13], the experiential tourism approach highlights the trend towards self-regulated and nature-based responsible tourism [14,15].

The stakeholders involved in tourism planning and management face various challenges to develop sound tourist solutions [16]. For this reason, it is important to understand the behavior patterns of tourists in a given area, which allows strengthening the tourism industry [17]. Actions such as configuring new businesses and thinking about new target groups [18], advertising attractions and tourist facilities [19]. These are commonly done to remove the main difficulties related to the choice of a tourist destination, such as the lack of information [8], the dependence on travel agents [20], the tour itinerary [21,22], and the low combination of interest points on a functional path [23]. Moreover, the discomfort caused by the travel experience is often associated with traffic in excess and environmental pollution around tourist areas [24]. Accordingly, the main variables used to plan a trip are related to the trip cost, time availability [25], reduction of vehicular traffic, and diversity of the landscape (bodies of water, mountains, and vegetation) [24], while the constraints were commonly found to be the queuing time, crowdedness, and weather [21,22,26].

Lately, technology has gained much attention, as it enables a link between information and communication technology and tourism [27,28]. Its use constitutes a path towards prosperity and development of countries and rural communities, which, until now, have only been visible through the spatial location of tourist attractions on the web [12,29]. Specifically, the use of computer software in the phase of planning and development of a tourist route requires the integration of features such as reliable information [16,30], the means of transportation and transportation infrastructure (public, private, and using bicycles or hiking), transportation time [31], optimal distance [32], available scenic attributes [33], and cultural heritage aspects [34]. In tourism, the use of “info-structure” computer tools has added many benefits to the value chain [35]. It contributes to user’s satisfaction, easing access to tourism service’s providers [2], increased employment, enabling recognition of tourist attractions, improving nature perception, sustaining product marketing [35], and developing the tourist routes and flows [9,36]. Typical examples of such computer applications are the use of geographic information systems (GIS) for spatial data analysis [37], monitoring, and planning of tourist destinations [38,39], as well as the evaluation of their potential alternatives [40]. The use of such technology has allowed different route patterns to be traced by integrating georeferenced data from the Internet [41], GPS, accelerometers, and other sensors incorporated in mobile devices or intelligent transport systems (vehicular) whose aim is to improve the tourist experience [42,43].

Depending on research objectives and facilities needed by the visitors to understand the environment, it is often necessary to construct maps that contain visual attributes such as infrastructure facilities, tourism network attractions [44,45], trails, and hierarchical routes, which make the core on tourist experience [46]. At this point, a route system increases legibility and graphic visibility of potential tourist activities [47,48] since it connects different types of features and resources such as the interest points, populated centers, historical infrastructure, cultural values, landscapes, restaurants, and accommodation facilities [49,50]. If located on a central route, these elements would significantly contribute to tourists’ development of authenticity and trip satisfaction feelings [51,52].

As tourist destinations, mountains have received a considerably increased number of visits in recent years, generating substantial economic benefits for local communities [53]. For instance, the Chimborazo Wildlife Reserve (CR) of Ecuador features the Chimborazo Volcano, which has been identified as one of the important tourist hotspots from the area [54,55,56]. However, there are many more tourist features in the area, while the general tourist knowledge regarding them, as well as their use, were found to be minor [55], indicating, therefore, either a lack of knowledge or a lack of sustainable connectivity of such features. Meanwhile, the CR attracts ca. 96,000 tourists per year, seeking mainly experiences related to high- and medium-mountain activities such as trekking, hiking, environmental interpretation, flora and fauna observation, and camping [57]. As a result, the location was found to be it is being crowding and unsustainable use of the local tourist resources. For these reasons, it is necessary to design and implement specific strategies to improve the environmental management in the CR and to enhance the value added by the tourist attraction [56]. There are many good strategies to do that, such as deploying a broader advertising campaign using regular channels, a better planning, and resource development by connectivity [58]. However, the sustainability of a tourist route should not be underestimated [59], since these routes are usually used at a group and intergroup level, being frequented by bicyclists, walkers and family trips in their own vehicles [52].

The aforementioned context has allowed us to determine the global importance of tourist routes for the strengthening of nature tourism. Ecuador, in general, and the CR, in particular, have several tourist attractions and facilities. However, the growing tourism demand in the CR is saturating the current routes and as a result decreasing the level of visitor satisfaction. It is important to join efforts to implement new tourist routes in this area and take advantage of resources that have not yet been exploited. In this sense, the objective of the study was to design an optimal nature tourism route in the Chimborazo Reserve. A series of network modeling tools incorporated in ArcGIS 10.5^®^ software were used to respond to this objective. Finally, the characteristics of the new route are related to the current route in economic aspects, time and distance.

## 2. Materials and Methods

### 2.1. Study Area

The Chimborazo Wildlife Reserve, created in 1986, is one of the sixty protected areas of Ecuador’s National System of Protected Areas [60]. It is located in three Andean provinces Chimborazo, Tungurahua, and Bolivar. With an altitude of 6268 m, it is the highest mountain in the country. The reserve is home to a great biodiversity, it is common to observe vicuñas, llamas, guanacos, rabbits, deer, wolves, and birds [61]. The San Juan rural parish is one of the 11 rural parishes of the Riobamba canton. It has been considered in this study because it represents the closest populated area of the Riobamba canton to Chimborazo Mountain. The whole study area features 17 tourist attractions that constitute important tourist hotspots for locals and tourists due to their natural and cultural beauty (Figure 1A,B).

This study was carried out based on the geographical and field-collected data of the CR, which is an Ecuadorian protected area featuring the Chimborazo Volcano. The volcano is characterized by an altitude of 6268 m above sea level [62], being located at the borders of the provinces of Chimborazo, Bolívar, and Tungurahua. Close to the CR there is a rural parish called San Juan, which is located in the Riobamba canton (Figure 1A).

### 2.2. Methodoly

In order to respond to the objective of the study, the methodology was divided into three sections. Initially, through a bibliographic review and field work, the interest points and tourist facilities in the study area were identified. Next, the design of the tourist route was developed using innovative office automation tools. Finally, a matrix was created to analyze the characteristics of the new route with the current route in economic aspects, time and distance.

#### 2.2.1. Identification of Interest Points and Tourist Facilities

In this first section, techniques such as literature review and field work were combined. For the identification of the interest points in the study area, the study conducted by Castillo et al. (2020) [56], was considered. Through field work between January and February 2021, all the interest points were contrasted and mapped by researchers from ESPOCH (Escuela Superior Politécnica de Chimborazo), to verify their correct functioning. In order to select the sites of interest, the preferences of tourists in recent years were considered, data that were obtained from visitors’ records and their comments. This information was also supported by the suggestions of the local guides when perceiving the satisfaction and preference of the tourist groups. Then, these points were included in a database in ArcGIS 10.5^®^ software. Subsequently, a coding procedure was developed for the identified points of interest to facilitate their representation in the study area. Finally, gray literature was used for the identification of tourist facilities, i.e., the San Juan Parish Land Use and Development Plan [63] and the CR Management Plan [64].

#### 2.2.2. Design of Efficient Routes

GIS-based network modeling was supported by the use of cost distance modeling (CDM), according to which the cost values grow with distance from a source point [65]. In the raster-based CDM, for instance, the cost values are cumulated with the distance from a point of origin, with the aim of generating a cost distance raster as a result. In other words, the procedure is based on the cell representation as nodes and links, where a node is the center of the cell, and a lateral or diagonal link connects the nodes to their adjacent cells. A side cell connects to its nearest neighbor cell, and this diagonal link connects the cell to one of the nearby neighbor cells. The cell distance is 1.0 for the link and 1.414 for the diagonal cell. The cost distance to travel from one cell to another through a lateral link is 1.0 defined by the average of the two cell cost values [66]:(1)$$1\times \frac{\left[\left(Ci+Cj\right)\right]}{2}$$
where
*Ci* = cost value of the cell *i*;*Cj* = the cost value of the neighbor cell *j.*

The cost distance was calculated using the Path Distance tool available in the ArcGis^®^ software, which calculates the distance by multiplying the surface distance, vertical factors, and horizontal factors. The surface distance is given by an elevation raster (DEM), and horizontal and vertical factors are inputs added in the model. The distance toolset of the spatial analysis tool in Arc Toolbox, gives the length of a line segment between the two points using the Pythagorean theorem (Euclidean direction) and the Euclidean allocation method. Method that assigns the cell, not data, the value at the closest source, whose objective is the identification of the closest proximity between two points. Additionally, the distance tool calculates the cost distance, cost back line [67]. The flow used to develop efficient routes is given in Figure 2.

The input files were obtained from the official governmental portals (Table 1), and they were projected to the Universal Transversal Mercator (UTM), World Geodetic System (1984), zone 17 S coordinate system, which is specific for the Continental Ecuador.

The tourist attractions were used to form a network of distances in a straight line to meet each other. Then, the field calculator tool of ArcGIS 10.5^®^ software was used to calculate the value of the length in kilometers between attractions in a straight line. In addition, a Triangular Irregular Network (TIN) of the Chimborazo mountain was also used. A TIN layer is commonly an elevation surface representing the height values across an area [71].

The initial and final points of the routes were set by taking as a starting point the SJ feature and the ending point the CB feature on route 1 (R1); the starting and ending points on the route 2 (R2) were the PF and CB features. It should be noted that point X (Figure 1) was common for both routes because it represents an important central point of information and tourist meeting operations in the CR. The selection criteria for the start and end points were formulated based on the study conducted by Castillo et al. (2020) [56], which points out that among the most frequented tourist attractions of the CR are the Chimborazo Mountain and the Polylepis Forest. Based on the initial and final points, two types of routes were drawn. The first route (red), which exists in the area, was designed for vehicular and bicycling traffic, and in some parts, it is used for trekking by tourists and locals. The second route (green), stands for the development of an efficient route as an alternative for tourists and locals to travel by bicycle and trekking in certain sections, depending on the level of difficulty and access. For spatial analysis, the initial, intermediary and final points were shapefiles created previously, standing for interest points. These points were used because they represent all the tourist attractions and tourist facilities used to carry on different tourist activities in the study area. Routes were developed based on the above-mentioned points using as an input the Digital Elevation Model (DEM) and shapefile (*.shp) of starting point SJ and the end point the CB in route 1 (R1) and between PF and CB in route 2 (R2) were used. The construction process had the following steps: (i) It began with loading the DEM (Figure 3a); (ii) Continued with the creation of a slope raster (Figure 3b); (iii) Finally, to draw the efficient route between the pre-established interest points, we generate three rasters, based on the criteria necessary to propose a route based on the ArcGIS 10.5^®^ software tool. The first was Cost Distance (Figure 3c), the second was Cost Back Link (Figure 3d) for which the starting point and the slope layer were used and the third Cost Path raster where the end point was selected, Cost Distance and Cost Back Link (Figure 3e,f).

In addition, in order for the tourist and the general population to be able to know and understand what the tourist potential of the study area is from the routes created based on the network of tourist attractions, a graphical proposal diagrammed to scale with relation to existing and proposed routes in ArcGIS 10.5^®^ software, through the Lucidchart (version 2020) [72].

#### 2.2.3. Creation of the Cost Matrix

Using a matrix table in Microsoft Excel, integrating elements were built such as code, distance, and time to and from interests points interesting in relation to X. In addition, the cost matrix has the state of the current and new road base on the features Shapefile [71]. To calculate the parameters of distance and time for both routes, the ArcGIS 10.5^®^ software field calculator tool was used to determine the values using by car, bicycle, and hiking. Azcarate (1984) [73] considered speed as a quantity that can be compared, measured, and expressed by numbers. In addition to being represented by a segment, it can be seen as a ratio of change of distance with time [74]. This information was validated with field trips to the current (red) and proposed (green) routes for three months (October to December 2020). The validation consisted of actual field trips with national and foreign tourists and researchers from the Escuela Superior Politécnica de Chimborazo (ESPOCH). In this process, the values corresponding to distance and time between the routes analyzed in the study were checked [75].

Finally, to calculate the share of efficiency between the red and green routes in both cases (routes 1 and 2) related to the estimated travel time, the total of the travel time of the red and green routes was carried out only for the case of cycling and hiking. This particular because routes 1 and 2 is in not designed for vehicles in ArcGis^®^ software. Then, the difference of the obtained values was calculated. Subsequently, the simple proportional reasoning method (rule of three) was applied, where having three known numbers: a, b, and c, such that, (a/b = c/x), (that is, a:b: c:x) the unknown number is calculated [76]. Based on this proportion, the share of efficiency of the estimated travel time was determined for cycling and hiking in CR. All the tasks related to analysis were carried out in Microsoft Excel (version 2020). The same software was used to produce the graphics needed in this study.

## 3. Results

For a better understanding, the results are presented in order, with respect to the objectives and methodology used. The first section presents the network of tourist attractions in the Chimborazo Reserve, describing the attractions and facilities in the study area. Next, the mapping of the current route and the newly designed route is detailed. Finally, a cost matrix is used to compare the routes in terms of distance and time.

### 3.1. Network of Tourist Attractions

In terms of interest points, 17 tourist attractions have been identified (Table 2), including landscapes such as mountains, rock formations, caves, ancestral resources, forests, archeological sites, mines, bodies of water, and viewpoints. Interconnected by sections that are consolidated with facilities such as roads, lodging and food services, shelters and tourist information. These attractions together with the facilities provide key tourism resources for the design of a nature tourism route.

All the tourist attractions described (Table 2), according to their geographic location, give rise to the establishment of a network with possible interconnections (Figure 4). The travel times were established considering the distances in a straight line and the mode used according to the geo-environmental conditions of the route. Among the attractions that the most stand out because they represent the focal points in the creation of the routes were the point two represented the Chimborazo Mountain, point eight Polylepis Forest. For instance, from point 2 to 8 and 2 to 17, there are 10.6 and 15.0 km, respectively. This fact represents the Euclidean distance between the initial and final points in both routes. In addition, there are points such as: 2(HM), 4(WN), 5(CR), 6(MT), 9(CH), 10(CC), 13(YR), 15(SU), 16(UP), and 17(SC), with a short distance in the network between them. This fact open possibilities to use most of the attractions in the design of these routes.

### 3.2. Mapping of Tourist Routes

Figure 5 and Figure 6 show the tourist routes 1 and 2. The red lines represent the current existing routes, while the green lines stand for the proposed routes. The blue lines represent the network of attractions and, the TIN denotes the Chimborazo mountain in the study area (Figure 4). For instance, considering the 2 routes, there are 5 strategic points: SJ, CB, PF, CB, and X, which were the start, middle, and end points of the routes in both tracks.

The current route 1 (red color) and proposed (green color) is shown in Figure 5A. The actual layout of the route was presented on the ground with 3 focal points: initial (SJ), centric (X), and final (CB). There are sections on the red and green routes that used the same spatial arrangement on the ground, due to the topographic conditions of the area. The red route represents the current existing stretch from SJ to CB, which is a paved road to X (Figure 1). Currently, tourists and locals travel along this road through vehicles and bicycles according to their interests. The green route represents the proposal to only for hike and bicycle. This route is 26% shorter than the current route at the CR (Table 3). This route provides new alternatives for tourists to take tours in this sector.

For a better understanding and comprehensive look at all the tourist facilities of the red and green routes, a proposal graphic of these routes is presented (Figure 5B). This route integrates 17 tourist attractions, five lodging centers, four food centers and one health center. The tour can be done by vehicle, bicycle, and hiking according to the interest points, whether for the red or green line.

Figure 6A shows the existing route 2 (red color) and proposed (green color). The real layout of the route is presented on the ground with 3 focal points: initial (PF), central (X), and final (CB). There are very few sections of the routes: red and green, which use the same spatial arrangement on the ground, due to the topographic conditions of the area. The red route represents the current existing section from PF to CB. This section is a second-order road, except for the road (E492) that crosses through X, which is paved route (Figure 1). Currently, tourists and locals travel through this road by means of vehicles, bicycles and hikes according to their interests. The green route represents the proposal to only for hike and bicycle. This new route is 43% shorter than the current route at the CR (Table 4). This fact provides new alternatives for tourists to take tours in this sector.

To better and have a comprehensive look at all the red and green route’s tourist facilities, a proposal graphic of these route is presented (Figure 6B). These routes include 11 tourist attractions, two lodging and food centers and one health center. The tour can be done by vehicle, bicycle, and hike according to the tourist’s interest, whether for the red or green line.

### 3.3. Cost Matrix

In the final section of the results to make cost comparisons considering time and distance between the current route and the new route design (Table 3). It is necessary to present a general description of the distances and approximate times of the stretches in the Chimborazo Reserve, with respect to point X. In addition, information is presented on the average speed depending on the mode of transportation and the condition of the road section.

As for the tourist facilities, there are two service providers: CL and CU, which are quite close by the existing and proposed routes. Finally, an important fact is also to point out that the distance and time of arrival at the closest health center to this route in relation to X is 16 km in 33 min by car for both routes. Additionally, Table 3 and Table 4 show important facts to highlight. There are sections where they cannot be done by vehicle or bicycle. For example, in the case of route 1: YR, MT, and WN, and for route 2: WN, MT, and YR, the only way to get there from X is by hiking, due to the topographic conditions and access roads to the place. Moreover, access to the infrastructure and tourist facilities can be carried out by the three forms of travel for both routes.

The Table 5 shows the shares of efficiency the routes related to the estimated travel time. In all the cases in both routes 1 and 2, by cycle and hiking, the green routes were more efficient compared with the red route. For instance, by hiking the green routes are 23.4% and 28.0% more efficient than red routes, in the routes 1 and 2, respectively.

Finally, the estimated travel time of the routes 1 and 2 by vehicle, cycling and hiking (Figure 7). For the case by vehicle, only the red route was represented because the green routes were designed to travel only by vehicle. On the other hand, by cycling and hiking, the trend was the same, which means that the green routes were more efficient related to estimated time compared with the red routes.

## 4. Discussion

People’s need to escape from routine has led individuals to seek different alternatives to relax and have fun [77]. Thus, it is essential to value the different resources that a territory has to develop various tourist activities [78]. In addition, the transport is a vital element in tourism as an enabling element that facilitates tourism activity, where the transportation is a strategic priority to improve the links between recreational opportunities [79]. Currently, to visit the CR the tourists have alternatives to travel by vehicle, cycling, and hiking. On the other hand, GIS has been widely used in both the public and private sectors to appreciate tourism with efficient transportations forms [80]. However, route mapping is a useful but little-used technique in visitor research, at least in the CR in Ecuador. In relation to the above-mentioned, this study showed new route alternatives through network analysis in GIS for tourists and locals visiting the CR.

In CR, several tourist attractions attract the attention of local residents and visitors in general [54,56]. Chimborazo Mountain and Polylepis Forest are among the most frequented attractions in the CR by locals and tourists, according to a study carried out by Castillo et al. (2020) [56]. These results provided us with the criteria for determining the focal interest points. Additionally, the study’s topographic provides optimal conditions for developing new routes, and tourists who may prefer detours or alternate routes to visit certain places and enjoy tourism to those that currently exist. In this context, it might be interesting to point out a study by Dickinson et al. (2004) [81] and Eaton and Holding (1996) [82] where they have reviewed the problems that the vehicles can cause to tourist trips concerning congestion and the impact on the behavior of trips in National Parks. That is why cycling and trekking have a big alternative for developing tourist activity [83]. In the case of the CR, there are very few vehicular and cycling routes for tourism development. Therefore, they need to be improved and adapted as an alternative optimal to travel. In many territories, this kind of “cycling” tourist transport has been encouraged so that cars’ user and tourist resources can be minimized [84]. For the tourist, the idea of traveling a safer, more feasible route in the middle of nature is very exciting [85]. As such, this study showed, due to their topographic and landscape conditions, through of ArcGis^®^ software. The green routes fit with these new trends, with non-motorized paths adapted to be done on hiking or on a bicycle. It is essential to have hiking and biking trails to enhance the tourist experience [86].

Another key behavior of the routes taken into study was calculate the approximate distance and estimated time of the tourist attractions and services of the existing and proposed routes for cycling and hiking in the CR. This fact should be analyzed with caution because in a research carried out by Alivand and Hochmair (2015) [87] they studied almost 100 routes selected by tourists and found that the travel time was approximately 90% longer than the fastest route chosen for them. Thus, the results reported in this study could be very important for visitors to choose alternative routes for cycling and hiking for a better enjoyment of tourist attractions and facilities in CR. On the other hand, hiking trails could be considered less detrimental to the daily life of the community in general and may limit some of the possible negative social and environmental impacts [88]. With this, environmentally friendly management could bring conservation benefits and enhance the tourism experience.

The tourism sector has grown rapidly in recent years. Consequently, the carbon footprint of this activity has also increased. For this reason, the World Tourism Organization (UNWTO) has opted for sustainable tourism as a central axis in the establishment of new tourism projects, where outdoor activities such as nature walks, climbing, photography, wildlife observation and paragliding prevail [1,89,90]. Tourism contributes 5% of the world’s carbon footprint. Of this, 3.2% comes from the transportation sector and the rest from the hotel industry [3,14]. This study proposes the design and implementation of a new nature route in the mountains, which will undoubtedly contribute to the reduction of the carbon footprint, since these activities generate a minimum impact and a low consumption of fossil fuels.

## 5. Conclusions

This study brings evidence on the creation of alternate routes for cycling and trekking for tourists in the CR. The route one includes 17 tourist attractions, five lodging establishments, four food centers and one health center. The route two integrates 11 tourist attractions, two accommodation and food establishments and one health center. These two trends generate alternatives for routes by vehicle, bicycle and hiking on the red routes and cycling and hiking on the green routes.

The contribution of this research was to maximize tour satisfaction by presenting in the green routes of visiting to tourist attractions, due to the growing demand in the CR. These two new green routes are shorter than the current routes related to approximate distance and estimated time at the CR. In addition to documenting such trends, this study’s results opened new ways to increase the possibilities of tours for the enjoyment of visitors in the study area.

Worldwide, the new preferences of tourism in a post-COVID19 situation point to an increase in the number of visitors to nature destinations. Considering this new reality, this study contributes to the strengthening of this tourism segment in the Chimborazo Reserve by providing an optimal design for the establishment of a new tourist route. Actions that allow us to be prepared as a destination of excellence in a post-pandemic situation and increase resilience to a greater number of visitors. These new strategies strengthen tourism management and satisfy a greater demand from tourists with respect to biosecurity. Finally, the main limitations in the research were due to the limited bibliographic information available and management problems in the research process due to the current pandemic situation.

## Figures and Tables

**Figure 1 ijerph-18-05293-f001:**
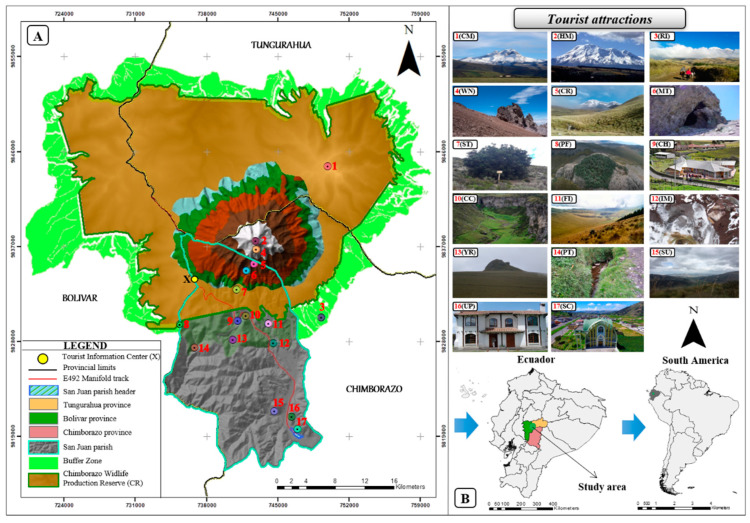
Map of the study area. (**A**) map of the CR and San Juan parish showing the location of tourist attractions; (**B**) location of the CR in Ecuador and South America.

**Figure 2 ijerph-18-05293-f002:**
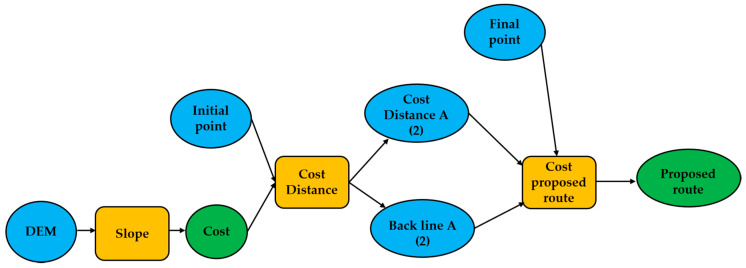
Visual programing language of Cost Distance modeling using Model Builder of ESRI.

**Figure 3 ijerph-18-05293-f003:**
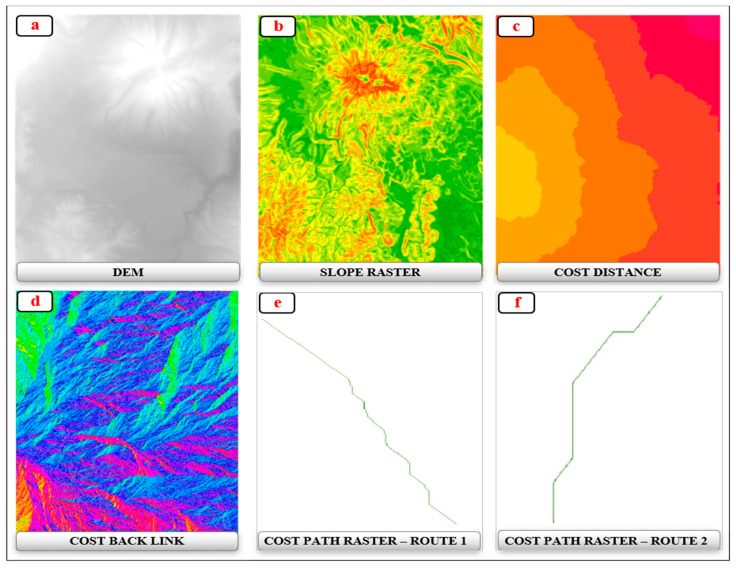
Raster obtained and best routes after running the cost distance algorithm in ArcGis 10.5. letter: (**a**–**d**) are rasters, while letters: (**e**,**f**) are the best routes.

**Figure 4 ijerph-18-05293-f004:**
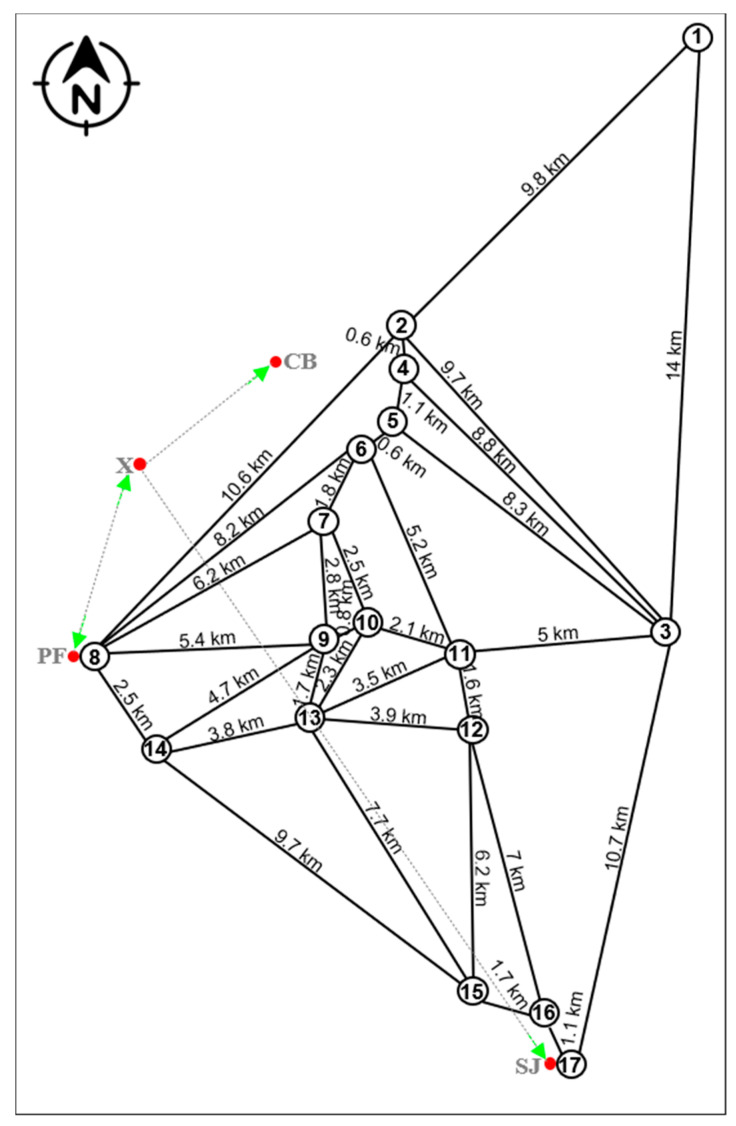
Network of tourist attractions in the study area.

**Figure 5 ijerph-18-05293-f005:**
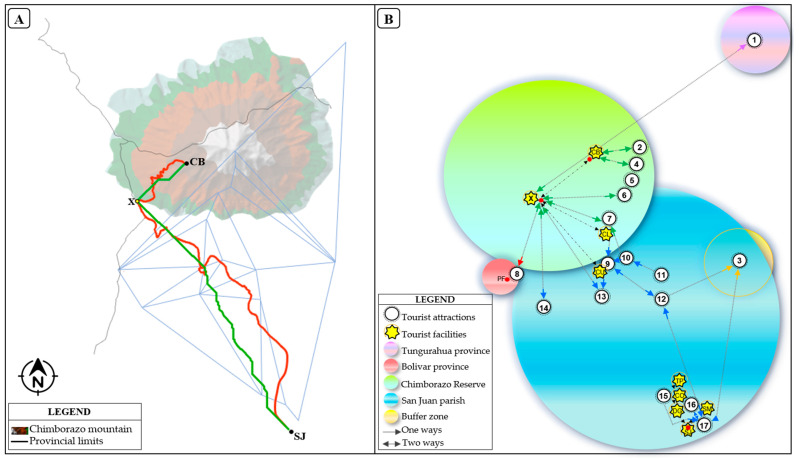
Route model 1. (**A**) current route (red color) and proposed route (green color); (**B**) graphic proposal of the red and green routes integrating all the tourist facilities in both routes.

**Figure 6 ijerph-18-05293-f006:**
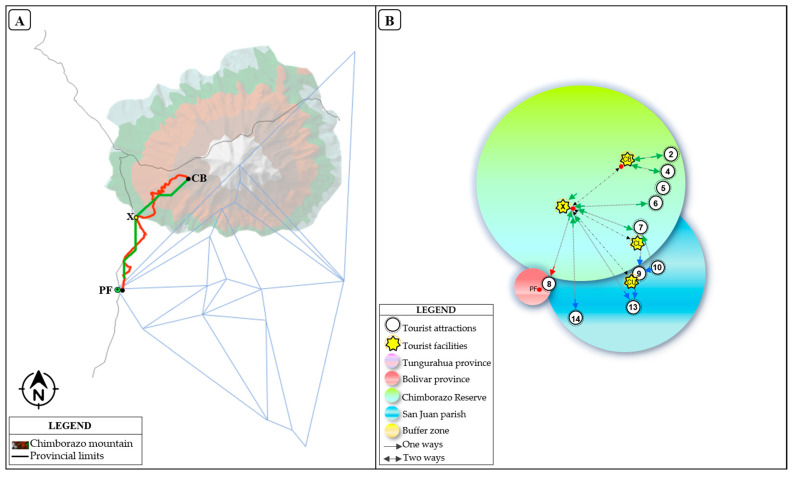
Route model 2. (**A**) current route (red color) and proposed route (green color); (**B**) graphic proposal of the red and green routes integrating all the tourist facilities in both routes.

**Figure 7 ijerph-18-05293-f007:**
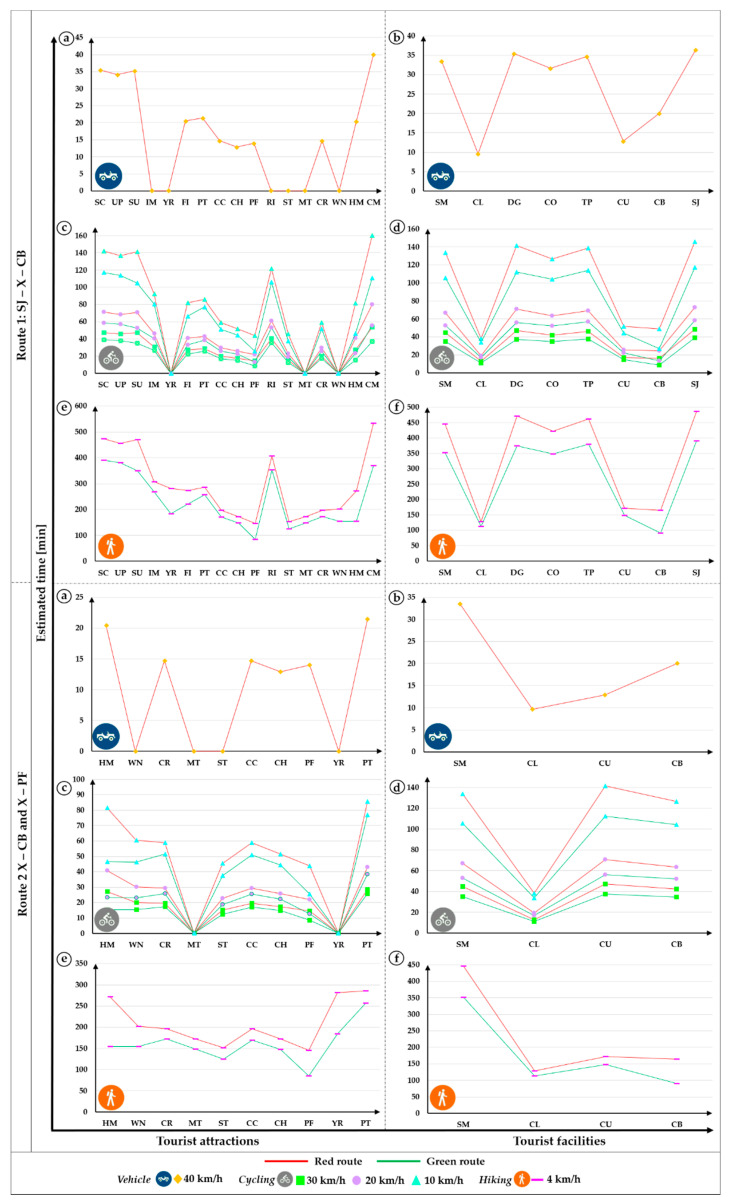
Estimated travel time by vehicle, cycling and hiking: Route 1, Legend (**a**,**c**,**e**) tourist attractions: SC, UP, SU, IM, YR, FI, PT, CC, CH, PF, RI, ST, MT, CR, WN, HM, and CM; (**b**,**d**,**f**) tourist facilities: SM, CL, DG, CO, TP, CU, CB, and SJ. Route 2, Legend (**a**,**c**,**e**) tourist attractions: HM, WN, CR, MT, ST, CC, CH, PF, YR, and PT; (**b**,**d**,**f**) tourist facilities: SM, CL, CU, and CB. The current route is represented in red color. Proposed route is represented in green color.

**Table 1 ijerph-18-05293-t001:** Input files used for the design of routes.

Input Files	Format	Source	Specifications
Digital Elevation Model (DEM)	Digital Elevation Model (*.dem)	Layers of geographic information *	DEM 30_Set 4 (30 Mb), UTM longitude from 864000 to 936000. 30 m Resolution per pixel.
			Projected Coordinate System: WGS_1984_UTM_Zone_17S
			Projection: Transverse Mercator
Triangular Irregular Network (TIN)—Chimborazo Mountain	Triangular Irregular Network (*.adf)	Layers of geographic information *	TIN: delaunay conforming
			Number of Data Nodes: 248474
			Number of Data Triangles: 492940
			Z Range: (3967.650391, 6240.000000)
			Projected Coordinate System: WGS_1984_UTM_Zone_17S
			Projection: Transverse Mercator
UTM coordinates of tourist attractions	Excel spreadsheet (*.xlsx)	Environment Ministry of Ecuador (2017) [57], Castillo et al. (2019) [54], Google maps (2020) [68]	Version 2020
Main road	Shapefile (*.shp)	Layers of geographic information *	Geometry Type: polyline
			Projected Coordinate System: WGS_1984_UTM_Zone_17S
			Projection: Transverse Mercator
Health center	Shapefile (*.shp)	National Information System of Ecuador (2014) [69], Google maps (2020) [68]	Geometry Type: point
			Scale 1:50.000
			Projected Coordinate System: WGS_1984_UTM_Zone_17S
			Projection: Transverse Mercator
Tourist facilities	Excel spreadsheet (*.xlsx)	Document from the review in Google maps (2020) [68]	Version 2020

Note: * Based on Information from Military Geographical Institute of Ecuador [70].

**Table 2 ijerph-18-05293-t002:** Interest points and tourist features in the study area.

No.	Interest Points	Short Description	Code	Province of Location		
				Chimborazo	Bolívar	Tungurahua
	Tourist attractions					
1	Carihuairazo Mountain	Three peaked volcano: It is a mountain glacier with an extension of rock to the top that is over 5020 m.a.s.l. It is allowed to go trekking, climbing and hiking activities.	CM			x
2	Chimborazo Mountain	Highest mountain/volcano in Ecuador: Highest mountain/volcano in Ecuador with 6268 m.a.s.l, known as the closest point to the sun. The main activities that can be carried out are: high mountains, observation of flora and fauna, trekking, photography, camping and research.	HM	x		
3	Route of the Ice Makers	Route of the ice makers: It is a millenary route by which the locals travel until they reach the ice mines.	RI	x		
4	Whympers’s Needles	It is a rock formation, it is named after Edward Whymper, the first conqueror of Chimborazo mountain.	WN	x		
5	Chimborazo Wildlife Reserve	Protected area: It is a Protected area of Ecuador, it represents one of the most important tourist attractions of the Province and the Ecuador, such as the Chimborazo mountain.	CR	x		
6	Machay Temple	Sacred cave of volcanic material: Sacred cave of volcanic material, where the ancient indigenous people of the area used it as a ceremonial and veneration center for the Chimborazo mountain.	MT	x		
7	Solitary Tree	Tree with ancestral values: It is a kind of large bush 5 m high and 6 m in diameter, which is located in the middle of the moor, surrounded by mounds of sand called dunes located on the slopes of Chimborazo mountain.	ST	x		
8	Polylepis Forest	Polylepis relict forest: It is a remnant of Polylepis forest, surrounded by a rock formation, represent a viewpoint to Chimborazo mountain.	PF	x	x	
9	Condor House	Community tourism center: Natural site where there are branches of forest, grasslands, pads, chuquiragua, among others. In this place there is a waterfall of about 30 m approximately.	CH	x		
10	Chorrera Canyon	Rocky formation: It is a rock formation in which a fall of crystalline waters of approximately 25 m high descends. The site presents rock formations that form an ideal wall for those who like climbing.	CC	x		
11	Fortress of the Incas	Archeological site: Ceremonial center, where it was considered that the Incas performed their rituals.	FI	x		
12	Ice Mines	Fossil ice mine: It represents the fossilized ice mine that is under 30 cm from the ground surface.	IM	x		
13	Yana Rumi	Black stone: Rock formation and ceremonial site for the locals.	YR	x		
14	Pogyo Tayta Andrés	Water spring: Formerly it was a healing center, people bathed in this place to cure their illnesses. Sacred rituals are performed thanking Pacha Mama for her kindness.	PT	x		
15	Shobol Urcu vantage point	Natural viewpoint: It is a mountain, constituted as a natural viewpoint, where you can see the Chimborazo mountain.	SU	x		
16	Union of Peasant Organizations of San Juan	Community tourism organization: Community tourism organization that offer tourist activities.	UP	x		
17	San Juan Bautista Church	Principal church: Main church of the San Juan parish. Locals venerate the image of San Juan Bautista.	SC	x		
	Tourist facilities					
1	Riobamba—Chimborazo E492 Manifold track	Paved road, main axis	DC	x	x	x
2	Shobolpamba Medical Dispensary	Health center located in San Juan Parish	SM	x		
3	Chimborazo Lodge	Lodging and food services—Private	CL	x		
4	Dream Garden Lodge	Lodging and food services—Private	DG	x		
5	Chimborazo country house	Lodging services—Private	CO	x		
6	Tambo Pak Samay	Lodging and food services—Private	TP	x		
7	Condor House Lodge	Lodging and food services—Community	CU	x		
8	Tourist Information Center	CR Tourist Information Center	X	x		
9	Carrel Brothers Refuge	First refuge of the CR. Lodging and food services—Private	CB	x		
10	San Juan parish	Town center and parish head	SJ	x		

**Table 3 ijerph-18-05293-t003:** General description of approximate distances and time of tourist routes in CR.

No.	Code	Approximate Distance of Interest Points in Relation to X (km) in Real Route		Estimated Time of Interest Points in Relation to X (min)									State of the Road **	
				Vehicle	Cycling						Hiking			
					Professional		Medium		Beginner					
				40 km/h *	30 km/h *		20 km/h *		10 km/h *		4 km/h at Average Pace *			
	Route 1: SJ-X-CB/Distance: 32.5 km (red), 24.0 km (green)													
	Tourist Attractions													
		**Red**	**Green**	**Red**	**Red**	**Green**	**Red**	**Green**	**Red**	**Green**	**Red**	**Green**	**Red**	**Green**
17	SC	23.7	19.5	36	47	39	71	59	142	117	474	390	pr	pr
16	UP	22.8	19.0	34	46	38	68	57	137	114	456	380	pr	pr
15	SU	23.5	17.5	35	47	35	71	52	141	105	470	349	pr/dr	pr/dr
12	IM	15.4	13.4	–	31	27	46	40	92	80	308	268	pr/dr	dr
13	YR	14.1	9.2	–	–	–	–	–	–	–	282	184	pr/dr	dr
11	FI	13.7	11.1	21	27	22	41	33	82	66	274	221	pr/dr	dr
14	PT	14.3	12.8	21	29	26	43	39	86	77	286	257	pr/dr	dr
10	CC	9.8	8.5	15	20	17	29	26	59	51	196	170	pr	pr
9	CH	8.6	7.4	13	17	15	26	22	52	44	172	148	pr	pr
8	PF	7.3	4.3	14	15	9	22	13	44	26	146	85	pr/dr	dr
3	RI	20.3	17.7	–	41	35	61	53	122	106	406	353	pr/dr	dr
7	ST	7.6	6.2	–	15	12	23	19	46	37	152	125	pr/dr	dr
6	MT	8.6	7.4	–	–	–	–	–	–	–	172	149	pr/dr	dr
5	CR	9.8	8.6	15	20	17	29	26	59	52	196	172	dr	dr
4	WN	10.1	7.7	–	–	–	–	–	–	–	202	154	dr	dr
2	HM	13.6	7.8	20	27	16	41	23	82	47	272	155	dr	dr
1	CM	26.7	18.5	40	53	37	80	55	160	111	534	370	dr	dr
	Tourist Facilities													
2	SM	22.3	17.6	33	45	35	67	53	134	106	446	352	pr	pr
3	CL	6.4	5.7	10	13	11	19	17	38	34	128	113	pr/dr	dr
4	DG	23.6	18.7	35	47	37	71	56	142	112	472	374	pr	pr
5	CO	21.1	17.4	32	42	35	63	52	127	104	422	348	pr/dr	dr
6	TP	23.1	19.0	35	46	38	69	57	139	114	462	380	pr	pr
7	CU	8.6	7.4	13	17	15	26	22	52	44	172	148	pr	pr
9	CB	8.2	4.5	20	16	9	25	14	49	27	164	90	dr	dr
10	SJ	24.3	19.5	36	49	39	73	59	146	117	486	390	pr	pr

Note: * Speed, ** Based on the features Shapefile [70]: paved road (pr); dirt road (dr).

**Table 4 ijerph-18-05293-t004:** General description of approximate distances and time of tourist routes in CR.

No.	Code	Approximate Distance of Interest Points in Relation to X (km) in Real Route		Estimated Time of Interest Points in Relation to X (min)									State of the Road **	
				Vehicle	Cycling						Hiking			
					Professional		Medium		Beginner					
				40 km/h *	30 km/h *		20 km/h *		10 km/h *		4 km/h at Average Pace *			
	Route 2: X-CB and X-PF/Distance: 15.5 km (red), 8.8 km (green)													
	Tourist Attractions													
		**Red**	**Green**	**Red**	**Red**	**Green**	**Red**	**Green**	**Red**	**Green**	**Red**	**Green**	**Red**	**Green**
2	HM	13.6	7.8	20	27	16	41	23	82	47	272	155	dr	dr
4	WN	10.1	7.7	–	20	15	30	23	61	46	202	154	dr	dr
5	CR	9.8	8.6	15	20	17	29	26	59	52	196	172	dr	dr
6	MT	8.6	7.4	–	–	–	–	–	–	–	172	149	pr/dr	dr
7	ST	7.6	6.2	–	15	12	23	19	46	37	152	125	pr/dr	dr
10	CC	9.8	8.5	15	20	17	29	26	59	51	196	170	pr	pr
9	CH	8.6	7.4	13	17	15	26	22	52	44	172	148	pr	pr
8	PF	7.3	4.3	14	15	9	22	13	44	26	146	85	pr/dr	dr
13	YR	12.2	9.2	–	–	–	–	–	–	–	244	184	pr/dr	dr
14	PT	14.3	12.8	21	29	26	43	39	86	77	286	257	pr/dr	dr
	Tourist facilities													
2	SM	22.3	17.6	33	45	35	67	53	134	106	446	352	pr	pr
3	CL	6.4	5.7	10	13	11	19	17	38	34	128	113	pr/dr	dr
7	CU	8.6	7.4	13	17	15	26	22	52	44	172	148	pr	pr
9	CB	8.2	4.5	20	16	9	25	14	49	27	164	90	dr	dr

Note: * Speed, ** Based on the features Shapefile [70]: paved road (pr); dirt road (dr).

**Table 5 ijerph-18-05293-t005:** Shares of efficiency between red and green route related to the estimated travel time.

Estimated Travel Time (min)					
	Vehicle	Cycling			Hiking
		Professional	Medium	Beginner	
Route 1: SJ-X-CB					
Red route total	264	434	651	1303	4998
Green route total	–	344	516	1033	3930
Share of efficiency (%)	–	23.3	23.2	23.1	23.4
Route 2: X-CB and X-PF					
Red route total	98	162	243	487	2076
Green route total	–	127	190	380	1599
Share of efficiency (%)	–	28.4	28.4	28.4	28.0

## Data Availability

All the data generated and analyzed during this study are included in this published article.

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
