# Peer review of "Design of Nature Tourism Route in Chimborazo Wildlife Reserve, Ecuador"

_ijerph, 2021, doi:10.3390/ijerph18105293_

Round 1

Reviewer 1 Report

The Science in the paper sounds robust and well.  Please check the following few issues for revision :

  • Figure 2: Please revise and correct it is supposed to be Cost Distance  and also Back line A please space and capitalize to ensure consistency
  • Check and revise line 242 something is missing 
  • line 271 check reading and flow
  •  

It would have been nice to have a discussion on how the routes can assist in the carbon reduction for the destination. 

Author Response

Dear Reviewer:

Your comments and corrections have been implemented. Please see attached the report of changes done.

Reviewer 2 Report

The aim of the study is a relevant  topic and would provide an excellent contribution to the growing literature in this area. The article provides results of original study which contributes new knowledges and experience in the field of tourism development.

Methodology  

Related to the first phase of the study “a bibliographic review and field work” it will the important to show how the points of interest and tourist facilities in the study area were identified, which inclusion or exclusion criteria was used and how, if so, the community participated in this phase. If not, why the community wasn´t integrated?

Results

This section should specify the characteristics of the points of interest and tourist facilities to improve the routes information

Discussion

This section misses some rather pertinent and impactful literature and contributions in this space that would significantly improve the manuscript as for exemple DüzgüneÅŸ E, Demirel Ö. 2013. Determining the tourism potential of the Altındere Valley National Park (Trabzon/ Turkey) with respect to its conservation value. Int J Sust Dev World Ecol. 20:358–368.

The topic and structure of this study as well as the quality of its presentation looks apropriate for the requirements of this Journal. I recomend to accept this article after minor revisions.

Author Response

(The authors gave the same response as above.)

Reviewer 3 Report

This article is very interesting that aims to design new routes as a specific strategy to improve tourism management and to increase the attractiveness of landscape features, promoting activities as a part of sustainable development. 
The new preferences of tourism in a post-COVID19 situation point to an increase in the number of visitors to nature destinations. Tourist interest points were identified and mapped using spatial analysis software(ArcGIS 10.5 ®), then two routes for bicycles and hiking were defined as being the most efficient, based on the most frequented tourist attractions. 
This study contributes to the strengthening of this tourism segment in the Chimborazo Wildlife Reserve(CR) by providing an optimal design for the establishment of a new tourist route. 
The research literature, data analysis, and methodology of this article are good. This tourist route of sustainable development may be provided to other countries for reference.

Author Response

Dear reviewer:

Thank you for your comments are gratifying.
